# Cost-Effectiveness of Strategies Addressing Environmental Noise: A Systematic Literature Review

**DOI:** 10.3390/ijerph22050803

**Published:** 2025-05-21

**Authors:** Nick Verhaeghe, Bo Vandenbulcke, Max Lelie, Lieven Annemans, Steven Simoens, Koen Putman

**Affiliations:** 1Department of Public Health, Interuniversity Centre for Health Economics Research (i-CHER), Vrije Universiteit Brussel, Laarbeeklaan 103, 1090 Jette, Belgium; bo.vandenbulcke@vub.be (B.V.); max.lelie@vub.be (M.L.); koen.putman@vub.be (K.P.); 2Department of Public Health and Primary Care, Interuniversity Centre for Health Economics Research (i-CHER), Ghent University, Corneel Heymanslaan 10, 9000 Ghent, Belgium; lieven.annemans@ugent.be; 3Department of Pharmaceutical and Pharmacological Sciences, KU Leuven, ON2 Herestraat 49-Box 424, 3000 Leuven, Belgium; steven.simoens@kuleuven.be

**Keywords:** cost-effectiveness, cost–benefit, cost-utility, environmental noise, systematic review

## Abstract

Environmental noise, a significant public health concern, is associated with adverse health effects, including cardiovascular diseases, cognitive impairments, and psychological distress. Noise reduction strategies are essential for mitigating these effects. Despite evidence of their health benefits, limited information exists on the cost-effectiveness of such strategies to guide resource allocation. This study systematically reviewed economic evaluation studies of interventions aimed at reducing environmental noise to assess their cost-effectiveness and inform policymaking. A systematic review following PRISMA 2020 guidelines was conducted across MEDLINE, EMBASE, and Web of Science. Eligible studies were full economic evaluations addressing environmental noise reduction strategies, assessing both costs and health effects. Screening and data extraction were performed independently by two reviewers. Quality appraisal employed the CHEERS 2022 checklist. Narrative synthesis was used to analyze findings due to heterogeneity in study designs, methodologies, and outcomes. Costs were standardized to 2024 euros. From 2906 identified records, five studies met the inclusion criteria, primarily focused on traffic-related noise. Three studies conducted cost-utility analyses, and two employed cost–benefit analyses. Reported interventions included sound insulation, take-off trajectory adjustments, and noise barriers. Economic evaluations varied significantly in methodologies, cost categories, and health outcomes. The health economic studies yielded mixed results, ranging from findings that demonstrated cost-effectiveness to those where the costs exceeded the benefits. There are currently too few health economic evaluations to draw robust conclusions about the cost-effectiveness of environmental noise mitigation strategies. Future research should adopt standardized approaches and robust sensitivity analyses to enhance evidence quality, enabling informed policy and resource allocation decisions.

## 1. Introduction

Environmental or community noise occurs in various contexts, including noise from transportation, leisure, housing, work, or industry, and generally refers to all unwanted, undesirable, or harmful sounds [1]. The World Health Organization defines environmental or community noise as ‘noise from all sources, except noise at the industrial workplace’. The main sources of community noise are road, rail, and air traffic, industries, construction and public works, and the neighborhood’ [2]. The principles of acoustics relevant to environmental noise focus on how sound is generated, transmitted, and perceived by individuals. Environmental noise is typically measured in terms of sound pressure levels, expressed in decibels (dB), and often adjusted using the A-weighting scale (dB (A)) to reflect human perception of noise. Key factors influencing sound propagation include distance attenuation, atmospheric absorption, and ground surface [3,4]. Low-frequency sounds, common in road [5] and rail traffic [6], tend to travel further and penetrate buildings more effectively. Standard noise metrics such as the equivalent continuous sound level (Leq), day–evening–night level (Lden), and nighttime level (Lnight) are used to assess exposure [4]. Understanding these acoustic principles is essential for evaluating the impact of environmental noise and for designing effective mitigation strategies.

Environmental noise has significant negative effects on both physical and psychological health, including cardiovascular diseases, hypertension, behavioral problems and cognitive impairments in children, annoyance, sleep disturbances, depression and anxiety, suicide [1,2,7]. For example, an umbrella review of systematic reviews and meta-analyses found that high noise exposure from various sources was linked to a 34% higher risk of cardiovascular disease and a 12% increase in cardiovascular mortality. It was also associated with elevated blood pressure (58–72%), diabetes (23%), and adverse reproductive outcomes (22–43%). Furthermore, the dose–response relationship indicated that the risks of diabetes, ischemic heart disease (IHD), cardiovascular mortality, stroke, anxiety, and depression rise with greater noise exposure [8].

Environmental noise also has an economic impact. Environmental noise affects various areas, including property values [9]. Brandt and Maennig [10] examined the impact of traffic noise on apartment sales prices in Hamburg, Germany, and found a reduction of approximately 0.23%. The negative impact of noise pollution on house values was also highlighted in the literature review by Yao et al. [11]. Conversely, the installation of noise barriers can lead to an increase in house prices [12]. In their study, Yao et al. [11] emphasized that this effect is location-dependent, with a price premium observed for houses situated closer to the core of the Montreal metropolitan area. A study conducted in South Korea employed the contingent valuation method to assess how much residents of eight metropolitan cities were willing to pay for noise reduction. It was found that the willingness-to-pay increased as the level of annoyance experienced increased [13]. Environmental noise, such as traffic, often leads to the need for noise barriers. These come with significant construction costs, including installation and ongoing maintenance expenses [14]. Environmental noise also has a significant health economic impact both in terms of direct medical costs and indirect costs due to its association with health problems. Direct medical costs are expenses directly related to medical care (e.g., hospital admissions, medication). Indirect costs are those arising from health issues that affect the ability to work or be productive [15,16]. In the European Union (EU), road transport, inland waterway transport, and rail transport together account for costs of EUR 64 billion related to healthcare and inconvenience for the population. Additionally, air traffic at a selection of 33 airports in the EU is responsible for an extra EUR 0.84 billion [17]. A study in the U.S. estimated that approximately 23.6 million working-age individuals (18–64 years) experience hearing loss. According to their research, people with hearing loss were 2.5 times more likely to be unemployed compared to those with normal hearing, and the average salary of workers with hearing loss was about 25% lower than that of workers without hearing loss [18].

Reducing exposure to environmental noise and mitigating its effects through noise management measures is therefore important for both individual health and public health. Noise mitigation in the transport sector involves a range of strategies aimed at reducing environmental noise. Various measures have been implemented to reduce transport-attributable noise, particularly in urban and suburban settings. These include constructing noise barriers along highways, and planting of hedges and shelter belts along roads, which are effective in lowering noise levels in suburban areas [9,19]. Other common strategies involve eliminating road and train horns in or near residential neighborhoods, relocating railway tracks [20,21,22], and enforcing vehicle noise limits to control emissions at the source [23].

A systematic literature review of the health effects of 43 noise interventions related to transportation demonstrated that many of the interventions were associated with positive changes in health outcomes, regardless of the type of source (road traffic, rail traffic, and air traffic), the outcome (annoyance, sleep disturbance, cardiovascular effects, cognitive development in children), or the type of intervention [9]. However, evidence regarding the effectiveness of noise management strategies on health is insufficient for policymaking. Budgets are limited, leading (health) systems to face the challenge of prioritizing the allocation of (financial) resources to strategies [10]. Such information can be obtained from health economic evaluation studies that provide payers and governments with evidence-based insights on how to spend available resources in the most efficient manner. In the context of noise mitigation, cost-effectiveness analysis compares the costs of different interventions relative to a specific outcome, such as the reduction in decibels (dBA) or decrease in the percentage of highly annoyed individuals. Cost–utility analysis evaluates interventions based on costs per quality-adjusted life year (QALY) gained, thereby integrating both the quality and quantity of life affected by noise exposure. Cost–benefit analysis monetizes both costs and benefits, allowing for a direct comparison—e.g., assigning monetary values to improved health, reduced annoyance, or increased property values. Cost-minimization analysis is applied when outcomes of different interventions are assumed or shown to be equivalent, focusing solely on identifying the least costly option [24].

To the best of our knowledge, there are no published systematic literature reviews that provide a summary of economic evaluation studies of interventions aimed at reducing environmental noise. The aim of the current study was to examine the published scientific evidence on the cost-effectiveness of strategies aimed at reducing environmental noise in society.

## 2. Materials and Methods

The review has been prepared and written in accordance with the ‘Preferred Reporting Items for Systematic Reviews and Meta-analyses (PRISMA)—2020 version’ [25]. A literature search strategy was developed for MEDLINE (via PubMed), and adapted for EMBASE (via embase.com), and Web of Science Core Collection (via Web Of Science) (Appendix A). The key concepts ‘environmental noise’, ‘strategies’, and ‘full economic evaluations’ were translated into search strings. For the first two key concepts, search strings were developed based on a preliminary search through Google to identify key words. For the concept of ‘full economic evaluations’, the search algorithm was based on a published systematic review on the cost-effectiveness of exercise referral schemes [26] and updated as, for example, ‘Cost-effectiveness analysis’ became a MeSH term in 2023.

Eligibility criteria were defined a priori for study selection (Table 1). The PICO (i.e., Population, Intervention, Comparator and Outcome) strategy was applied to describe the criteria. To be eligible for inclusion, studies had to focus on individuals exposed to environmental noise (‘Population’ criterion), as there is scientific evidence demonstrating the exposure-response relationship for a number of health effects [27]. The intervention could include all strategies to reduce noise in the community, both indoors and outdoors. Studies that focused on strategies to reduce noise pollution other than in the community (e.g., earphones) were outside the scope of the literature review. Only ‘full’ economic evaluation studies, i.e., studies that examined and reported both the (incremental) costs and (incremental) health effects of at least two alternative strategies, were eligible for inclusion [24]. No a priori restrictions were placed on how health effects were expressed. This meant that both health effects expressed in ‘natural units’ (e.g., avoided cardiovascular disease) and ‘utilities’ (health-related quality of life) were eligible for inclusion [24]. During a preliminary exploration of the literature, we identified a number of cost–benefit analyses without reporting (in detail) the impact of a strategy on health costs and effects in the analysis. As we are interested in the impact of strategies to mitigate environmental noise on health (and costs), such studies were not eligible for inclusion. There was no lower limit on the publication date and studies could be included until 31 December 2023.

Two reviewers (N.V. and B.V) independently screened the titles and abstracts yielded by the search, blinded to each other’s decision, using the web application Rayyan [28]. Disagreement about inclusion or exclusion was resolved by discussion, otherwise, a third reviewer was consulted. Selection of the records was based on the eligibility criteria (Table 1). Second, screening on full text was executed by one reviewer (N.V.), against the same eligibility criteria. During the second screening round, reasons for exclusion were noted (Appendix A). A data extraction form was developed in Microsoft Excel^®^ based on a published template [29]. The following information was extracted from each included study: general study characteristics (e.g., publication year, country, noise mitigation strategy, noise source), methodological approach (i.e., model-based or within-trial) and characteristics (e.g., time horizon, perspective, cost categories, health outcomes), results, and author’s conclusion. The data extraction was performed by one author (NV). The principal outcome measures were health economic outcomes being (incremental) health outcomes, (incremental) cost outcomes and incremental cost-effectiveness ratios (ICERs), incremental cost-utility ratios (ICURs), and cost–benefit ratios.

The reporting quality of the included studies was assessed using the ‘Consolidated Health Economic Evaluation Reporting Standards 2022 (CHEERS)’ checklist [30]. This checklist contains 28 items and associated recommendations on the minimum amount of information that should be included when reporting economic evaluations. A score ‘1’ was assigned if the item was reported and a score ‘0’ if the item was not reported. If an item was not applicable, the score ‘NA’ was assigned. Studies were not excluded after the quality assessment and the results of the quality assessment are reported at the individual study level. A narrative synthesis of the results was performed because of the limited comparability of the included studies. Indeed, they differed widely in design, methodology or outcome measures. The findings of each study were analyzed and described in detail, taking into account similarities and differences in outcomes, context, and methodological quality. In this way, a nuanced overview of the available knowledge could be obtained despite the diversity in the studies. Different currencies were converted to euros (reference year: 2024; reference country: Belgium).

## 3. Results

### 3.1. Study Selection

Initially, 2906 records were identified. After removing the duplicates, 2227 records remained which, based on the inclusion and exclusion criteria, were screened for title and abstract. After this selection, 22 records remained of which, for 2 records, the ‘full text’ (after contacting the first and last author) could not be obtained. The full texts of the remaining 20 records were evaluated for inclusion and exclusion criteria, after which five studies were finally included in the literature review (Figure 1).

### 3.2. General Study Characteristics

Most of the studies were conducted in the US [31,32,33] (Table 2). All studies were an economic evaluation of interventions aimed at reducing traffic-related environmental noise (Table 2). In two studies, the economic evaluation consisted of a cost–benefit analysis [34,35], while the research design in two studies [31,33] was a cost–utility analysis and in one study [32] a cost–effectiveness analysis (Table 2).

### 3.3. Methodological Charcteristics

The economic evaluation was conducted in two studies [31,33] using a decisional Markov model (Table 3). The perspective from which the analysis was performed was not reported in three studies [32,34,35]. In the two studies [31,33], where the perspective was reported, the analysis was conducted from the societal perspective (Table 3). The types of costs included differed from study to study. Zafari et al. [33], Jiao et al. [31], and Wolfe et al. [32] examined both the impact of the strategy aimed at reducing environmental noise on direct medical costs and indirect costs. In Wolfe et al. [32], also the willingness to pay per dB of excess noise was considered. Lopez et al. [34] referred to ‘costs associated with health and well-being’ including improvements to living standards at home, work and recreational environment. Prendergast and Staff [35] referred only to ‘intervention costs’. The studies also differed in the type of health outcome taken into account (Table 3). The time horizon was ‘lifetime’ in three studies [31,32,33]. In the study by Lopez et al. [34], the time horizon was not reported, while in the study by Prendergast and Staff [35] it was presumably one night (Table 3). In three studies [31,32,33] with time horizons exceeding one year, costs and effects were discounted. In contrast, Lopez et al. [34] only discounted costs. Three studies [31,32,33] examined the impact of the environmental noise intervention on costs and health effects for specific conditions, while in the Lopez et al. [34] study this was described as the impact on ‘total health’ (Table 3). In two studies [34,35], sensitivity analyses were not performed to check uncertainty related to the input values of key parameters.

### 3.4. Results Economic Evalution Studies

The study by Lopez et al. [34], which investigated the effect of measures to reduce noise caused by gantry cranes, found that the total investment cost of the measures was higher (year 2022, CAD $2,786,448 [year 2024, EUR 1,931,200]) than the health and welfare benefits for people living near the port. This resulted in a return-on-investment of −40% (Table 4).

### 3.5. Reporting Quality Appraisal

Zafari et al. [33] investigated the impact of changing the take-off trajectory for aircraft on costs and health effects for cardiovascular disease and anxiety among local residents of LaGuardia Airport, New York. The authors concluded that limited use of 1 particular take-off trajectory compared to continuous use of this trajectory was cost-effective (ICUR: USD 10,006/QALY [EUR 10,301/QALY]). The probabilistic sensitivity analysis showed that 25% of simulations were dominant. At a willingness to pay of USD 50,000/QALY and USD 100,000/QALY, 75% and 85% of simulations were likely to be cost-effective, respectively. Varying the base case value of the input parameter ‘relative risk of anxiety disorder’ had the most effect on the outcome (ICUR) (Table 4).

Jiao et al. [31] examined the impact of installing sound insulation (aimed at decreasing dB exposure) in homes near LaGuardia Airport, New York, on costs and health effects on cardiovascular disease and anxiety disorders. The intervention was found to be more expensive than the current situation (incremental cost: USD 6793 [EUR 6993/QALY]) but yielded greater health gains (incremental QALYs: 0.61). This resulted in an ICUR of USD 11,163/QALY [EUR 11,492]. The sensitivity analyses showed that varying the value of the input parameter ‘relative risk of anxiety disorder’ had the most effect on the ICUR. The probabilistic sensitivity analysis showed that, at a willingness to pay of USD 50,000/QALY, installing sound insulation was a cost-effective intervention in 91% of the simulations (Table 4).

Prendergast and Staff [35] examined the impact of installing noise barriers along the railway tracks on sleep problems among local residents. The median probability of awakening at least one time each night decreased from 95.2% to 88.0%. The median probability of being awoken three or five times a night decreased from 50.9% to 29.4% and 9.2% to 2.7%, respectively. The analysis showed a total noise benefit of 106–152 dB per 1 million Australian dollars at a total cost of AUD 1.8–2.1 million. (Table 4). This was considered as cost-effective as it exceeded a threshold of 100 dB benefit per AUD 1 million.

Wolfe et al. [32] examined the cost–benefit (cost per 1 dB decrease) of sound insulation in homes and purchasing homes and land by 16 airports in the US. The results showed that at an income level of USD 40,000/year, the benefits ranged from $0 per person at 0 dB removed to USD 19,000/person at 35 dB removed. At 20 dB DNL avoided in areas with average annual income levels of USD 40,000 per person or at 16 dB DNL in areas with average annual income levels of USD 60,000 per person, the insulation cost was on average covered by the benefits; For income levels up to USD 40,000, welfare benefits from willingness-to-pay (WTP) never exceeded the cost of even the lowest cost land acquisition. At 65 dB DNL, noise insulation projects were USD 7000 per person higher than the benefits. However, at 75 DNL dB, the cost of home insulation was equal to the welfare benefits. For land acquisition policies, the welfare benefits were USD 30,000 less than airport land acquisition costs (Table 4).

Table 5 summarizes the reporting quality of the five included studies based on the CHEERS checklist. Overall scores ranged from 9/28 for the study by Prendergast and Staff [35] to 21/28 for the study by Jiao et al. [31]. Remarkably, no study reported details related to the study population such as demographic, socioeconomic or clinical characteristics. Only two studies [31,33] reported the perspective from which the analysis was performed, and in two studies [34,35], no sensitivity analyses were performed to investigate possible sources of uncertainty in the analyses.

## 4. Discussion

The aim of the literature review was to synthesize the published scientific evidence on the cost-effectiveness of strategies aimed at reducing environmental noise. On the one hand the findings highlight the growing recognition of the health and economic burden associated with environmental noise. While efforts have been made to quantify the health effects and economic implications of environmental noise mitigation strategies, the evidence on the cost-effectiveness of such interventions remains limited. The amount of literature on the effectiveness of strategies aimed at reducing environmental noise is much greater as showed by several literature reviews [3,36,37]. Aside from the limited health economic evidence, the reporting quality of the included studies in the review raises concerns. Many of the studies failed to provide key information necessary for a transparent and thorough assessment. Critical elements such as a clear description of the study population, the perspective or time horizon over which costs and outcomes were measured were often not reported. This lack of detail complicates correct interpretation of the results and makes it more difficult to adequately assess the methodological soundness of health economic evaluations. For example, three studies did not report the perspective of the analysis, which is critical for interpreting costs and benefits across stakeholders. Similarly, the absence of information on study populations characteristics limits the generalizability of the findings. This is worrying given the growing need for well-reported health economic evaluations to inform evidence-based decision-making in healthcare and environmental policy [30]. Without comprehensive and methodologically sound health economic evaluations, policymakers lack crucial data to prioritize environmental noise mitigation efforts and understand the potential healthcare savings and health improvements that could be achieved through targeted strategies aimed at reducing environmental noise. Ensuring that studies are of solid methodological quality is thus essential for their practical applicability and for drawing reliable conclusions that can inform decision-making.

The studies included in the review varied in terms of methodological approaches including differences in perspectives, time horizons, outcome measures, cost categories, environmental noise reduction strategies. This complicates comparing the outcomes of the studies and consequently the formulation of clear conclusions about the cost-effectiveness of environmental noise reduction strategies. Together with the fact that some studies lacked critical methodological information limited the interpretability of individual study results and introduced uncertainty in our synthesis. Additionally, the transferability of the findings to other contexts can be questioned given that contextual factors such as payment and reimbursement systems, economic climate, governmental and regulatory factors influence policy decision-making processes [38]. Context is particularly important in health economic evaluations of strategies aimed at addressing environmental noise. It can vary greatly depending on factors such as urban characteristics (e.g., population density, infrastructure) [39], neighborhood socio-economic status [40], type of environmental noise [41]. Another concern relates to the different approaches applied to calculate noise levels and exposure. Indeed, there are numerous methods for calculating noise levels at specific receiver locations, and the resulting data can be expressed using a wide range of noise indicators. In noise research, both of these factors pose significant challenges, as they greatly hinder the comparability of results across different studies [42,43].

While the overall number of health economic evaluations is limited, the included studies suggest that certain interventions, particularly those targeting noise insulation in residential areas may be cost-effective. For example, the study by Jiao et al. [31] evaluating sound insulation near LaGuardia airport reported a relatively low ICUR and high probabilities of cost-effectiveness. In contrast, infrastructure-level interventions to reduce noise caused by gantry cranes, found that the total investment cost of the measures was higher than the health and welfare benefits for people living near the port [34]. Policymakers should therefore prioritize investments in noise mitigation strategies that are both context-specific and supported by transparent, high-quality economic evaluations to ensure efficient allocation of resources and maximal health benefit.

We were interested in the impact of strategies aimed at reducing environmental noise on health outcomes (in addition to the impact on costs). During the papers’ selection process, several economic evaluations were identified not considering the impact on health outcomes (e.g., [44,45,46]). These studies were not considered for inclusion in the current literature review. In four of five studies, the strategy was aimed at addressing environmental noise caused by traffic. This is yet only one source of environmental noise, alongside noise from industry, construction and public works, and the neighborhood [2,47] with negative effects on health [1,9]. For example, the results of a cross-sectional study in Denmark of 3893 adult residents of apartment buildings found that noise pollution from neighbors was strongly associated with several somatic (e.g., headache) and mental (e.g., sleep problems, depression) health problems [48]. The acoustic environment as perceived by people is an important aspect for the comfort of people in general, as shown by environmental studies [49]. In order to assess the acoustic environment, both qualitative and quantitative methodologies are common such as soundwalks, listening tests, interviews, and focus groups [50]. For example, Barros et al. [19] conducted a socio-acoustic survey in three residential areas along a highway in the Flanders region of Belgium to assess the impact of newly installed noise barriers. While average daytime noise exposure at locations near the barriers decreased significantly by 4.4 to 11.7 dBA, direct reductions in noise annoyance levels were observed in only one of the three areas. Furthermore, it could not be concluded that the installation of noise barriers had any important effect on residents’ quality of life, health complaints, concentration difficulties, or sleep quality.

The literature review has a number of limitations. First, only a limited number of studies examining the cost-effectiveness of environmental noise reduction strategies were identified. Moreover, methodological heterogeneity across the studies was observed. This makes it impossible to draw robust conclusions about the cost-effectiveness of environmental noise reduction strategies. Another limitation concerns the likely publication bias with an increased possibility that interventions that did not prove to be cost-effective are less likely to be published. This bias may have skewed the evidence base included in our review, leading to an overrepresentation of interventions that appear cost-effective. Consequently, the overall conclusions regarding the economic value of environmental noise reduction strategies should be interpreted with caution. Although we attempted to minimize this risk by conducting a comprehensive search across multiple databases, we acknowledge that the potential for publication bias cannot be entirely ruled out. Furthermore, only studies in English, French, or Dutch were considered for inclusion. It is therefore possible that we have missed relevant papers in another language.

## 5. Conclusions

In conclusion, the review showed that the evidence based on the cost-effectiveness of environmental noise mitigation strategies is limited in quantity as well as in quality of studies. Although exposure to environmental noise has been associated with several health issues and is increasingly recognized as a public health concern, economic analyses assessing the benefits of noise reduction are scarce. The included studies were heterogeneous in study design and outcome measures complicating comparison and synthesis. Moreover, important elements such as the perspective, study population characteristics, uncertainty analyses were often poorly reported, further limiting the interpretability and applicability of findings. This lack of information and transparency makes it difficult for policymakers to evaluate the (long-term) value-for-money of such interventions. Further research should adopt high-quality, transparent methodological approaches to enable policymakers to make informed decisions about investments in environmental noise reduction strategies. Strengthening the evidence base will enable more efficient resource allocation and support policy measures that improve public health.

## Figures and Tables

**Figure 1 ijerph-22-00803-f001:**
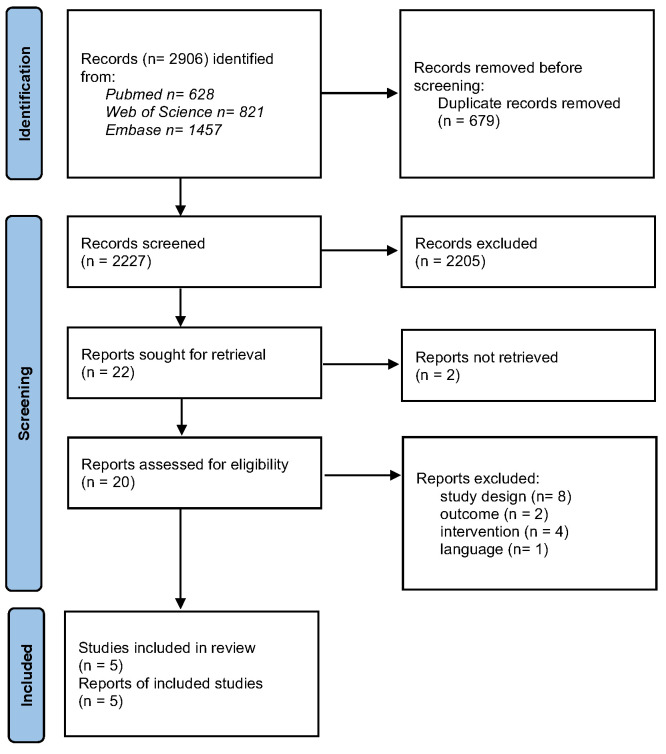
Study selection process.

**Table 1 ijerph-22-00803-t001:** Eligibility criteria.

	Inclusion Criteria	Exclusion Criteria
Population	Persons exposed to environmental noise	Persons exposed to noise other than environmental noise
Intervention	Strategies aimed at reducing environmental noise, both indoors and outdoors	Strategies aimed at reducing other forms of noise
Comparator	All comparators (e.g., no intervention, any other intervention)	/
Outcome	Cost–effectiveness, cost–utility, cost–benefit, cost-minimization	Cost–benefit evaluations without health outcomes considered
Study design	Cost–effectiveness analysis, cost–utility analysis, cost–benefit analysis, cost-minimization analysis	Other study designs
Geography	No restrictions	/
Language	English, French, Dutch	Other languages

**Table 2 ijerph-22-00803-t002:** General characteristics of the studies included in the systematic review.

First Author (Publication Year)	Country	Study Design	Study Population	Environmental Noise	Strategy	Comparator
Lopez (2024) [34]	Canada	CBA	residents of Halifax harbor	road and rail noise from cranes	low-noise paving and lifting mechanisms, rail track polishing	no measures assuming a baseline sound level of 85 dB
Zafari (2018) [33]	US	CUA	population Queens, NY affected by aircraft noise (within 60 dB)—LaGuardia airport (*n* = 83,807)	aircraft noise	alternative take-off route	usually used take-off route
Prendergast (2017) [35]	Australia	CBA	20 residential properties within approx. 50 m of railway track	railway noise	noise mitigation including sound barriers and track lubrication system (over 6 km route)	no mitigation measures
Jiao (2017) [31]	US	CUA	hypothetical cohort (*n* = ?) near LaGurdia airport NY	aircraft noise	sound insulation in dwellings aimed at noise exposure <55 dB	current exposure >55–≤65 dB
Wolfe (2016) [32]	US	CEA	residents of 16 US airports	aircraft noise	noise insulation in homes (n = 10) + homes and land acquisition (n = 6) by 16 airports	no aviation noise reduction

Note: dB, decibel; CBA, cost–benefit analysis; CEA, cost–effectiveness analysis; CUA, cost–utility analysis; km, kilometer; NY, New York; US, United States.

**Table 3 ijerph-22-00803-t003:** Methodological characteristics of the studies included in the systematic review.

First Author	Model	Perspective	Time Horizon	Costs	Currency/Year	Outcome	Discount Rate	Sensitivity Analyses
Lopez [34]	/	not reported	not reported	intervention cost, health and well-being costs	CAD, 2022	Health and well-being	costs and effects: 4%	/
Zafari [33]	Markov	societal	life time	direct medical costs CVD and anxiety disorder, indirect costs (productivity loss), intervention cost	USD, 2016	QALY	costs: 3% effects: 3%	OWSA, PSA
Prendergast [35]	/	not reported	one night?	intervention cost (noise mitigation measures)	AUD not reported	Probability of noise-induced awakenings 1, 3 and 5 times/night -environmental sleep disorder	/	/
Jiao [31]	Markov	societal	life time	direct medical costs CVD and anxiety disorder, indirect costs (productivity loss), intervention (sound isolation) cost	USD, 2016	QALY	costs: 3% effects: 3%	OWSA, PSA
Wolfe [32]	/	not reported	life time	direct medical and indirect costs stroke, hypertension and myocardial infarction, WTP/dB of excess noise intervention cost	USD, 2010	changes in dB	costs: 3% effects: 3%	PSA

Note: dB, decibel; CVD, cardiovascular disease; OWSA, one-way sensitivity analysis; PSA, probabilistic sensitivity analysis; QALY, quality-adjusted life year; US, United States; WTP, willingness-to-pay.

**Table 4 ijerph-22-00803-t004:** Results of the studies included in the systematic review.

First Author	(Incremental) Costs	(Incremental) Effects	Result	Sensitivity Analyses
Lopez [34]	Health and well-being benefits CAD 4,127,870 [EUR 2,860,898] Investment costs: CAD 6,914,318 [EUR 4,792,098]	/	Investment costs CAD 2,786,448 [EUR 1,931,200] higher than benefits/ROI: −40%	/
Zafari [33]	Incremental cost USD 11,288 [EUR 11,621]	Incremental QALYs: 1.13	ICUR: USD 10,006/QALY [EUR 10,301]	alternative take-off route: WTP USD 0/QALY: 25% simulations cost-effective; WTP USD 50,000/QALY: 75%; WTP USD 100,000/QALY: 85% RR anxiety: most sensitive parameter
Prendergast [35]	/	Median reduction awake 1 times/night: 95.2% to 88.0%; 3 times/night: 50.9% to 29.4%; 5 times/night: 9.2% to 2.7%	Total sound benefit 106–152 dB/AUD 1 million at a total cost of AUD 1.8–2.1 million	/
Jiao [31]	Incremental cost USD 6793 [EUR 6993]	Incremental QALYs: 0.61	ICUR: USD 11,163/QALY [EUR 11,492]	RR anxiety: most sensitive parameter, most sensitive cost parameter: intervention costs; noise insulation cost-effective in 91% of simulations at WTP USD 50,000/QALY
Wolfe [32]	/	/	Income level USD 40,000: benefits: USD 0/person (−0 dB) to USD 19,000 (−35 dB); −20 dB DNL (income levels of USD 40,000/year) or—16 dB DNL (income levels of USD 60,000/year), insulation costs = benefits; Income levels of USD 40,000, welfare benefits (WTP) never exceed the cost of even the lowest cost land acquisition; at 65 dB DNL: noise insulation USD 7000/person higher than benefits; at 75 dB DNL: noise insulation costs = benefits. Land acquisition: welfare benefits USD 30,000 less than airport land acquisition costs	

dB, decibel; ICUR, incremental cost-utility ratio; QALY, quality-adjusted life year; ROI, return on investment; RR, relative risk; WTP, willingness-to-pay.

**Table 5 ijerph-22-00803-t005:** Assessment of reporting quality of included studies.

	Item	Lopez [34]	Zafari [33]	Prendergast [35]	Jiao [31]	Wolfe [32]
**1**	Title	0	0	0	0	0
**2**	Abstract	1	1	1	1	1
**3**	Background and objectives	1	1	1	1	1
**4**	Health economic analysis plan	0	0	0	0	0
**5**	Study population	0	0	0	0	0
**6**	Setting and location	1	1	1	1	1
**7**	Comparators	1	1	0	1	1
**8**	Perspective	0	1	0	1	0
**9**	Time horizon	0	1	0	1	1
**10**	Discount rate	1	0	0	1	1
**11**	Selection of outcomes	1	1	1	1	1
**12**	Measurement of outcomes	1	1	1	1	1
**13**	Valuation of outcomes	NA	1	NA	1	NA
**14**	Measurement and valuation of resources and costs	1	1	0	1	1
**15**	Currency, price date, and conversion	1	1	0	1	1
**16**	Rationale and description of model	NA	1	NA	1	NA
**17**	Analytics and assumptions	0	0	0	1	1
**18**	Characterizing heterogeneity	0	0	0	0	0
**19**	Characterizing distributional effects	0	1	0	1	0
**20**	Characterizing uncertainty	0	1	0	1	1
**21**	Approach to engagement with patients and others affected by the study	NA	NA	NA	NA	NA
**22**	Study parameters	1	1	0	1	0
**23**	Summary of main results	1	1	1	1	1
**24**	Effect of uncertainty	0	1	0	1	1
**25**	Effect of engagement with patients and others affected by the study	NA	NA	NA	NA	NA
**26**	Study findings, limitations, generalizability, and current knowledge	1	1	1	1	1
**27**	Source of funding	1	1	1	0	0
**28**	Conflicts of interest	1	1	1	1	0
**TOTAL**		14	20	9	21	15

## Data Availability

No new data were created or analyzed in this study. Data sharing is not applicable to this article.

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
