# Peer review of "Cost-Effectiveness of Strategies Addressing Environmental Noise: A Systematic Literature Review"

_ijerph, 2025, doi:10.3390/ijerph22050803_

Round 1

Reviewer 1 Report

Comments and Suggestions for Authors

Dear authors,
I have read with interest the manuscript entitled "Cost-effectiveness of strategies addressing environmental noise: A systematic literature review". I believe that you have studied an important issue in the field of environmental noise. The methodology used seems relevant to me, and even if only five articles were considered, the criteria were clearly established for conducting the literature review.

However, I must make some general comments before this manuscript can be accepted for publication. 

General comments

1. The authors do not demonstrate an understanding of the main principles of acoustics. 

2. The general costs of noise, from an economic point of view, are not sufficiently well established in the early sections of the manuscript. Several articles in the literature have emphasized, in particular, that noise can be a negative externality and result in economic impacts (see Cavallaro and Nocera, 2022).

3. Attention must be paid to the main noise mitigation methods used in the transport field, given the scope of the manuscript and the five articles identified. No information is presented in the manuscript on this aspect, even though it is a central theme of the analysis.

4. The economic impacts of noise mitigation measures must be considered in the manuscript (see Yao et al. 2021).

5. Given that the articles in the literature review relate to distinct noise sources and that mitigation measures are varied, it is very difficult to put forward elements that can be mobilized in public policy.

6. A critical appraisal of the quality of the selected articles must be carried out, and the particularities of local contexts must be better brought to the fore in the discussion.

7. The conclusion remains too simple and succinct. What needs to be put forward and why?

8. Please define the cost-effectivenness, cost-utility, cost-benefit and cost-minimization methods in the field of noise mitigation. 

9. Attention should be paid in the discussion to the importance of socio-acoustic surveys that assess the perception of the effects on quality of life and health associated with the implementation of mitigation measures in given environments. 

References

Cavallaro, F., & Nocera, S. (2022). Are transport policies and economic appraisal aligned in evaluating road externalities?. Transportation Research Part D: Transport and Environment, 106, 103266.

Yao, Y. B., Dubé, J., Carrier, M., & Des Rosiers, F. (2021). Investigating the economic impact of noise barriers on single-family housing markets. Transportation Research Part D: Transport and Environment97, 102945.

Reviewer 2 Report

Comments and Suggestions for Authors

Thank you for the opportunity to review your manuscript. The topic addressed is timely and important, particularly given the growing attention on the health and economic implications of environmental noise. Your systematic review is well-structured and fills a notable gap in the literature by focusing on full economic evaluations of noise mitigation strategies. Below are a few suggestions that may help improve the clarity, depth, and utility of the manuscript: consider broadening your search strategy to include studies published in other languages or grey literature sources (e.g., government reports, conference proceedings). This could strengthen the comprehensiveness of the review, given the small number of included studies. While the eligibility criteria are well presented, it would be helpful to include more detail on the reasons for exclusion during full-text screening. A supplementary table listing excluded studies and reasons for exclusion would add transparency. Some included studies lack critical methodological information (e.g., study population, perspective, time horizon). While this is a limitation of the primary studies, it would be valuable for the authors to explicitly discuss how these omissions affect the interpretation and generalizability of findings. The discussion could be enriched by reflecting more on the policy implications of the findings. For example, how can the results guide public investment in noise mitigation? Are certain types of interventions (e.g., sound insulation vs. infrastructure changes) more promising? While the heterogeneity among studies is acknowledged, if any data could be standardized (e.g., cost per QALY), consider including a simple quantitative summary or visual representation (e.g., forest plot or cost-effectiveness plane). The limitations section is appropriate, but could benefit from additional discussion on publication bias and the challenges of comparing interventions across diverse contexts (e.g., different countries, health systems, or noise sources).

Just a typographical edit: please check row 29 of paragraph 4-Conclusion. Additionally, consistent use of terms like “cost-effectiveness analysis” versus “economic evaluation” would improve clarity.

Round 2

Reviewer 1 Report

Comments and Suggestions for Authors

Dear authors,
Thank you for all your efforts in responding to my comments. The manuscript has been considerably improved. I feel that the responses are clear and complete. As the limits of the manuscript had already been established, I have no further comments of my own.